# A Qualitative Exploration of Policy, Institutional, and Social Misconceptions Faced by Individuals with Multiple Chemical Sensitivity

**DOI:** 10.3390/ijerph22091383

**Published:** 2025-09-04

**Authors:** Susan J. Yousufzai, Elaine Psaradellis, Rohini Peris, Caroline Barakat

**Affiliations:** 1Faculty of Health Sciences, Ontario Tech University, Oshawa, ON L1G0C5, Canada; caroline.barakat@ontariotechu.ca; 2St. George’s University School of Medicine, St. George’s University, True Blue Campus, St. George, Grenada; 3Association Pour la Santé Environnementale du Québec-Environmental Health Association of Québec (ASEQ-EHAQ), Saint Sauveur, QC J0R 1R1, Canada; elaine@aseq-ehaq.ca (E.P.); office@aseq-ehaq.ca (R.P.)

**Keywords:** air quality, personal care products, health policy, multiple chemical sensitivity, social misconceptions, volatile organic compounds

## Abstract

Multiple Chemical Sensitivity (MCS) is characterized by recurring symptoms in response to low-level chemical exposures that are typically well-tolerated by the general population. Despite the debilitating health impact of MCS, public indifference and prevailing skepticism often result in stigma, misinformation, and systemic barriers that obstruct individuals’ access to essential environments. This qualitative study examined the lived experiences of individuals with MCS, focusing on how their condition is misunderstood and the factors that contribute to misconceptions about MCS. Seven focus group transcripts were analysed using thematic analysis in NVivo. Participants (aged 50–60) were drawn from various regions in Canada. One main category emerged from the analysis, centred on misconceptions influenced by policy and community factors. This category was divided into four themes, each with subcategories: (1) Psychological misattribution of MCS, (2) Healthcare and Institutional Gaps, (3) Policy Barriers, Compliance, and Resistance, and (4) Commercial Influences and Misleading Practises. These themes suggest a need for improvements in policies and transparency related to chemicals used in household and personal-care products, institutional compliance with fragrance-free guidelines, and increased awareness of MCS to reduce stigma and misconceptions. Addressing these issues can lead to adequate accommodations and support systems, which significantly improve quality of life.

## 1. Introduction

Multiple Chemical Sensitivity (MCS) is a chronic condition characterized by heightened sensitivity to various chemicals present in everyday environments [1,2]. It is a condition that impacts a substantial portion of the population worldwide, with reported prevalence rates ranging from 1% to 33% [3,4]. In Canada, the most recent data suggest that 3.5% of individuals aged 12 years and older have been diagnosed with MCS [5,6]. That is over 1 million Canadians, who are predominantly women (72%) [6].

Despite the growing number of studies that focus on the prevalence and progression of MCS, there is still speculation regarding its etiology and underlying mechanisms [1,3,4,5]. Indeed, MCS has been considered an underrecognized disease due to its shared comorbidity pattern and multiple organ system effects, making it challenging to diagnose patients [5,7,8]. Although some diagnostic markers have been suggested, there is no known diagnostic marker for MCS. The current practise for diagnosis relies on patient history and self-reported symptoms [5,9,10]. In addition, the educational gap regarding MCS in medical schools, as well as into continuing health programs for health care providers in practise—suggested to be due to the difficulty of incorporating new material into the curricula—contributes to challenges in diagnosis and management of MCS [11,12].

Furthermore, there is a disparity in the recognition of MCS by various countries [4,13]. While several European countries, such as Germany, Luxembourg, and Austria, classify it under the International Classification of Diseases (ICD-10) as an “unspecified respiratory condition”, “unspecified allergies”, or “hypersensitivity”, in Italy, it is recognized at the local level as a rare disease [14]. Japan also acknowledges MCS, using the ICD-10 code T65.9 and J68.9 [4,15]. In Canada, MCS is recognized by federal and local agencies. However, The World Health Organization (WHO) has not assigned a separate ICD-10 code for MCS [4]. The 1999 consensus provides the most comprehensive case definition of MCS, as “a chronic condition with symptoms that recur reproducibly in response to low levels of exposure to multiple unrelated chemicals and improve or resolve when incitants are removed” [16] (p. 147). The 1999 consensus criteria have also been validated using a reproducible questionnaire [17], identifying four specific neurological symptoms (i.e., having a stronger sense of smell than others, feeling dull/groggy, feeling “spacey,” and having difficulty concentrating) to discriminate between most patients and controls [18]. Recent research recommends the Environmental Exposure and Sensitivity Inventory (EESI) and its shortened version, the quick EESI (QEESI), for screening MCS [19,20,21]. In addition, a three-item screening questionnaire, the Brief Environmental Exposure and Sensitivity Inventory (BREESI), has also been validated as a screening tool for chemical intolerance, with a recommendation to confirm with the QEESI [22]. These tools may help address the medical education, treatment, and diagnosis gap for MCS in practise among healthcare professionals.

Recent advancements in research have elucidated the cause-and-effect relationship of MCS, as a consequence of specific chemosensory receptors that are widely distributed throughout the central nervous system (CNS) becoming sensitized to low-level chemical exposures [12,23]. These sensory receptors are classified as the transient receptor potential (TRP) family, specifically the vanilloid 1 (TRPV1) and ankyrin 1 (TRPA1) subfamilies, and predominantly respond to chemical stimuli [23,24]. Repeated exposure to poor air quality is known to induce neurobiological changes, wherein the receptors become hyperexcitable and upregulated over time. Specifically, in MCS patients, Molot et al. [12] suggest that exposure to pollutants increases the number of these specific receptors. As these receptors increase and become more sensitized, people experience symptoms even at low dose exposures, thus lowering the threshold for which people may tolerate generally acceptable indoor and outdoor pollutants [7,12,25]. Furthermore, due to the widespread distribution of these sensory receptors within the CNS, and downstream signalling, hypersensitivity of the TRP receptors within the CNS can impact other organ systems, causing varying intensity of symptomology throughout the respiratory, digestive, endocrine, muscular, or cardiovascular systems. Thus, individuals affected by MCS can experience a wide range of symptoms and comorbidities, such as non-food allergies, arthritis/rheumatism, headaches, fatigue, respiratory difficulties, cognitive impairments, and gastrointestinal disturbances, even when exposed to low levels of certain chemicals [1,3,6].

Risk factors for MCS are also multifactorial and include genetic predispositions, the interplay between genetic and environmental factors, oxidative stress, inflammation, cell dysfunction, and psychosocial factors [12]. Furthermore, as a primary risk factor for MCS, these challenges are exacerbated by a lack of awareness of the ubiquitous pollutants that infiltrate both the outdoor and indoor environments [12]. Corroborating this understanding is, then, the evidence linking long-term exposure to air pollution as a risk factor for various non-communicable diseases [2,26]. This association is attributed to the disruption of cellular detoxification mechanisms, which typically mediate the hazardous effects of environmental stressors [12,27]. For instance, volatile organic compounds (VOCs) in essential and non-essential products used in indoor environments (fragrances, scented products, personal care, cleaning and laundry products, “deodorizers” and disinfectants, dry-cleaned clothes, furnishings, and building materials [12,28] have been linked to activation of the TRP subfamily [12]. Correspondingly, evidence suggests that MCS patients are generally more reactive when exposed to indoor environments that are saturated with low levels of VOCs emitted from household cleaning products [29]. VOCs sensed by TRPV1 receptors include chemicals such as m-xylene (present in paints, air fresheners), toluene (paint thinners, adhesives), styrene, benzene, ethylbenzene, acetone (present in nail polish removers), diethyl ether (stain removers), hexane, heptane, cyclohexane, and formaldehyde (used in disinfectants, air fresheners, as a preservative) [12,30,31,32]. Notably, a Canadian study analysing 84 VOCs in indoor environments identified nearly 50 different VOCs present in more than half of 3800 homes, with apartments exhibiting higher concentrations [33,34]. Since MCS is a chronic condition that may be exacerbated by exposure to VOCs, more attention must be given to the correlation between long-term consequences of exposure to VOCs and MCS [12,35], especially given that the independent, synergistic, or additive effects of VOCs found in various everyday household products often remain unknown until after being on the market for years.

The perception of having an intolerance to usually tolerated environmental exposures perpetuates indifferences between those with MCS and those without [12]. Indeed, research indicates that MCS is a complex, multifaceted chronic condition involving both neurological and physiological factors. Correspondingly, the physiological symptoms that manifest can have a negative impact on all aspects of an individual’s daily life, including employment, social interactions, educational pursuits, and overall well-being [36,37], emphasizing the need for greater recognition and support for those affected. For instance, Gibson et al. [38] found that MCS significantly impairs individuals’ ability to participate in daily activities and access healthcare services. Additionally, other researchers emphasized the pervasive stigma and discrimination experienced by individuals with MCS, further exacerbating their difficulties [36,39,40]. These findings underscore the urgent need for increased societal awareness and targeted policy interventions to address the systemic barriers faced by individuals living with MCS.

Misconceptions are ideas or beliefs held by a group of people that lack scientific validation [41]. Identifying misconceptions is essential in the context of health management, as they can significantly influence preventive and treatment measures related to chronic conditions that are not well understood. Misconceptions often emerge in situations where society is divided over the scientific validity of a health-related issue—a pattern commonly observed during disease outbreaks, such as the COVID-19 pandemic [41]. Similarly, misconceptions surrounding MCS have been the subject of research, highlighting their profound impacts on affected individuals. For example, beliefs that MCS is likely psychogenic [42] or that individuals are overreacting to environmental triggers undermine the legitimacy of their experiences and create barriers to effective care [4,36]. Addressing these misconceptions is critical for the management and prevention of MCS.

While existing studies have shed light on how individuals with MCS cope with a socially delegitimized medical condition and live within a context characterized by the constant presence of chemical products [25,36,37,38,39,40], research on the consequences of the illness from the perspective of those affected is relatively scarce [25]. Building on this perspective, this paper aims to explore the perceived misconceptions faced by individuals with MCS and how these misconceptions may create barriers to their daily lives, indirectly impacting their health outcomes. Specifically, this study aims to explore the policy and community-level misconceptions surrounding MCS and how these perceptions impact access to healthcare and social inclusion. By exploring the lived experiences of individuals with MCS, this research may help bridge the divide within the medical community regarding the legitimacy of the condition [7]. Additionally, the insights gained can inform the development of effective interventions and strategies to enhance the quality of life for those affected. This study has the potential to inform policies and practises that enhance support for individuals with MCS, while also contributing to efforts aimed at reducing stigma and discrimination. Ultimately, the findings aim to foster greater inclusivity and understanding for those living with MCS.

## 2. Materials and Methods

### 2.1. Participants

The participants in this study included individuals with MCS (*n* = 38), who were required to experience symptoms of MCS as the primary inclusion criterion. Although a formal diagnosis was not mandatory, the vast majority (81.6%) had an official diagnosis from a healthcare professional.

Prior to the commencement of the study, the purpose and procedure were explained to the interviewees, and they were given the opportunity to ask questions. The participants were informed that the recorded and transcribed focus groups would be treated confidentially and that their anonymity would be guaranteed in the presentation of the findings. All participants provided consent to partake in the focus groups and have their experiences shared. Ethics approval to conduct secondary analysis of the data was provided by Ontario Tech University (REB # 17951).

### 2.2. Data Collection

Participants were drawn from various regions across Canada. Recruitment was primarily conducted through email invitations sent to the membership base of the Association pour la santé environnementale du Québec-Environmental Health Association of Québec (ASEQ-EHAQ). The email outlined the focus group’s purpose, eligibility criteria, and confidentiality measures, emphasizing that all personal information would be de-identified. Additionally, the invitation highlighted how their participation would contribute to a better understanding of MCS, encouraging individuals already familiar with the condition to join and share valuable insights.

Focus groups were conducted by Association pour la santé environnementale du Québec- Environmental Health Association of Québec (ASEQ-EHAQ) with individuals who have MCS (either diagnosed or undiagnosed) on three separate occasions (November 2022, December 2023, and January 2024). In total, seven focus groups were conducted across the three occasions, concentrating on lived experiences of avoidance and prevention to address how they intersect with barriers to inclusion and access, and environmental exposure inequities (with particular attention to workplaces and housing). Furthermore, the focus group questions centred on understanding the support and information required by individuals and organizations to implement and adhere to fragrance- or scent-free policies effectively. Participants were also asked about the impact of avoidance on managing MCS, including the challenges they face, changes in products on the market such as scent boosters, and the symptoms they experience despite efforts to avoid triggers. Additionally, the questions explored access to medical treatment for MCS symptoms and effective prevention strategies. The questions asked were informed by a combination of previous research, direct reporting via phone or email from people with MCS to ASEQ-EHAQ, and feedback gathered through community discussions. This multi-faceted approach ensured that the questions addressed the real-world experiences of people with MCS, focusing on the issues that are most relevant and pressing for the community.

### 2.3. Data Analysis

Focus groups were recorded and transcribed verbatim. During the transcription of focus groups, any information that could potentially identify participants was removed from the transcript. To confirm this process, a review of all transcripts was conducted by one researcher to ensure that they were anonymized by replacing any identifiers with pseudonyms. All data were imported and organized into NVivo version 14 and analysed using Thematic analysis. Thematic analysis is a flexible qualitative research method that enables researchers to explore the perspectives of participants [43]. One researcher read a total of seven transcriptions several times to identify sentences or paragraphs with the same meaning through an open coding process, allowing for an in-depth exploration of participants’ experiences and perspectives. Initial codes were identified by significant phrases, words, or ideas.

Labels reflecting a higher level of abstraction were inductively assigned to each code under broader themes that represent patterns in the data. These emerging codes were reviewed and grouped into preliminary categories based on their similarity. The categories were further discussed with the research team to define the final themes and assigned concise names that reflect their meaning. This process enhanced consistency and credibility in the analysis process. The quotes shown in this article were selected due to their representativeness of the recurring experiences brought up by participants.

## 3. Results

### 3.1. Demographic Characteristics

Table 1 displays the demographic characteristics of participants. Most participants fell within the 50–60-year age bracket, and the majority identified as female (89.5%). In addition, more than half of the participants were from Ontario (52.6%) (Table 1).

### 3.2. Summary of Qualitative Findings

The analysis of focus group discussions with individuals living with MCS identified three primary categories: (1) Sources of Misconception, (2) Impacts of Misconception on daily life, and (3) Recommendations for improvement. This paper focuses exclusively on the first category—Perceived Policy-level and Community-level Factors Leading to Misconceptions, which is further divided into four distinct themes (Figure 1). A complementary manuscript will explore the remaining categories, delving into the impacts of misconceptions on daily life and providing detailed recommendations for addressing these challenges.

Figure 1 illustrates the frequency of shared experiences by participants corresponding to each theme and subtheme. In relation to theme 1, participants expressed the impact of misconceptions that perpetuate stigma and social bias regarding their condition as psychogenically misinterpreted. Subtheme 1.1, concerning stigma and social bias, was mentioned thirteen times. This subtheme consisted of experiences discussed by participants that involved facing dismissal when trying to educate others, as well as social bias that their physical symptoms from exposure to scented products were psychological. Subtheme 1.2, impact on individual identity, comprises six experiences mentioned by participants, involving a loss of their former selves and lives. They discuss having aspects of their life taken away from them or lost, such as their employment, friends, and family. As a consequence, multiple participants experience isolation and mental health impacts.

Furthermore, barriers to accessing appropriate treatment, care, and essential services (subtheme 1.3) were mentioned seventeen times. This subtheme comprised experiences from participants explaining the barriers that prevent access to critical services, such as healthcare environments, pharmacies, grocery stores, and housing. Barriers include the pervasive selling of scented products, which are not only used by people but also adhere to surfaces and other products, making it challenging to access these environments and essential needs. In addition, participants report exposure to chemicals and mould from substandard building materials.

In relation to subtheme 2.1, nine experiences were reported by participants related to receiving limited care and being dismissed by healthcare providers due to insufficient training and awareness of MCS among healthcare providers. This led to multiple participants seeking expensive alternative treatments. Subtheme 2.2, the role of government in shaping public perceptions of MCS, included experiences that highlighted the perpetuation of barriers due to the lack of implementation of fragrance-free products in public spaces, suggesting a lack of consideration given to people with MCS by government entities. This suggests that unless a change is implemented at multiple systematic levels, the lack of accommodations given for people with MCS by government entities will continue to add to the perception that MCS is not a serious health condition. Theme three, Policy Barriers, Compliance, and Resistance, consists of two subthemes. Subtheme 3.1, concerning gaps in addressing MCS in government policy, was mentioned by participants nine times as an issue. Participants expressed that there are limited regulations in place to implement policies across different legislations, at the federal, provincial, and municipal levels, due to a lack of understanding and consideration of MCS as a health condition. In addition, multiple participants shared experiences in which they observed resistance within institutions to implementing or adhering to a fragrance-free policy, despite the use of posters indicating a fragrance-free environment (subtheme 3.1). Participants suggested the need for stricter enforcement measures, more government involvement, and effective advertising.

Multiple participants agreed on the influence of commercial interest on product safety standards, noting experiences where they have felt there was misleading information from commercials (subtheme 4.1); this was mentioned eight times. Finally, many participants reported experiencing an increase in the intensity of scents in products, as well as the addition of fragrances to various products that did not previously contain them (subtheme 4.2). Participants suggest a link between this and COVID-19, particularly with the increased use of scented sanitizers.

### 3.3. Themes and Subthemes Representing Experiences of Participants

#### 3.3.1. Theme 1: Psychological Misattribution of MCS

Subtheme 1.1: Related stigma and social bias

Individuals with MCS frequently face dismissal due to misconceptions that their condition is psychological and their symptoms are exaggerated. Participants reported that their experiences are often invalidated until the physical consequences and manifestations of MCS become evident. Many shared that they are frequently told their symptoms “must be in their head,” implying a psychological basis rather than a legitimate physiological condition with real symptoms and consequences. This recurring dismissal, coupled with negative interactions when discussing MCS, leaves participants feeling denigrated and marginalized. The stigma surrounding MCS is often rooted in its lack of universal recognition as a medical diagnosis and the subjective or difficult-to-quantify nature of its symptoms and triggers, such as reactions to everyday chemicals. These factors add to the difficulty of educating others.


*One of the things that is a benefit for me is that I have a tactile response to the chemicals. So you’ll see it in my face. […] and when it’s severe, I’ll have it in my throat and probably end up with an asthma attack where I can’t breathe. So that actually works because then people actually see this. But for the most part, when I talk about it, people like to deny it and dismiss it and denigrate it, denigrate me. It’s all in my head. And of course it is. That’s where my nervous system is, by the way.*



*Yes, my family is very respectful, but sometimes, perhaps more so in my entourage, they still think that it’s psychological, that it’s stress, they don’t understand the biological side.*


These experiences illustrate how the invisibility of certain symptoms and the tendency to misattribute the condition to mental health contribute to the stigma and invalidation faced by individuals with MCS. This reinforces feelings of isolation and misunderstanding, even within supportive environments.

Subtheme 1.2: Impact on individual identity

Participants expressed the emotional toll of losing key aspects of their previous identities, such as employment, friends, and family, leading to feelings of isolation, frustration, and a diminished sense of independence. These experiences underscore the broader social and psychological consequences of widespread misconceptions about MCS.


*The other side of that coin is that before all of this happened to me, I would have been considered an extrovert. I love people. I was an educator, had fun, all that kind of stuff. I am definitely now using my introvert side to get along to the point where people call me a hermit and I come back at them and say, I’m a hermit, just to make things clear. And so when you end up removing yourself from life, from activities, how I am labelled, how I would be a different person.*



*So my avoidance has become isolating. When I go to a grocery store, I am always with someone else because if I’m triggered, I need to leave immediately. I get anaphylactic and my throat swells. It has really affected my independence*


Subtheme 1.3: Related barriers to accessing appropriate treatment, care, and essential services.

Participants reported that when seeking support from health care providers, their condition is often misinterpreted as psychogenic and thus treatable through mental health interventions. This misunderstanding discourages individuals from pursuing such assistance, as they feel their condition extends beyond a mental health issue and encompasses significant physiological dimensions.


*But I’ve also taken on a hospital. I had to sign a gag order that I did take on the hospital and a doctor with success a number of years ago through human rights. But during that process, you know the dismissiveness of trying to make me look as though I was mentally ill, marginalizing me, just telling me to go elsewhere, and creating a huge unit of people to try to take me down.*



*There’s always something in the environment that’s going to affect me. It was like winter. In winter, it’s often the chimney fires when I take my walks outside. I can feel it. […]. It’s not in my head. It’s not psychological. I don’t need to go to a psychiatrist. It’s really physical. I’ve had this condition for 7 years. […] Some people in the medical world who say it’s in your head, that’s really insulting.*



*And I nearly died from this at least a couple of times. From chemicals or chemical exposure, smoke, perfume, medications, even that were given to me that are wrong. Even the ones that are right, made me sicker. And it’s just mental what it did to me. I didn’t go down the psychiatric route, because how many doctors wanted to put me into psychiatry? And I said no, because I feel in my brain, I’m okay.*


Participants highlighted the challenges created by a consumer culture that normalizes the widespread use of fragranced personal and household products. This cultural norm often forces people with MCS to advocate for their health needs when asked to use or tolerate products that trigger serious physical symptoms. They emphasized the contradiction in a society that promotes inclusivity for different groups while marginalizing those with MCS. This exclusion reveals a perceived hierarchy of needs, where the health and accommodations of people with MCS are regularly ignored or dismissed.


*At every doctor’s office I’ve been, they have a sign saying it’s scent free, but then they have hand sanitizer next to it, and they ask you to use it. So I’ve spoken up about this, but nobody gets it. So that’s extremely frustrating. Because this hand sanitizer makes me very sick, but they’re usually very accommodating when I tell them I can’t use it. And let me just go to the bathroom and wash my hands.*



*The consequences are appalling for all of us, whether it’s for a day or a month. It’s just not normal. That it should be triggered, in any case, in essential service institutions. So we can do without going to a play, we can do without a family party, we can even have Zoom tomorrow, then have Zoom family parties, then have online fun. But the lack of access to essential services is a form of abuse, period. When you don’t offer, then when it’s a society that calls itself inclusive, we accept everyone, no matter what, drag, queer, you know, well, okay. Except us, our little gang. Well, not you guys, for example. We won’t accommodate you. We won’t give you the easy way out. We’ll keep telling you that it’s all in your head and that you’re imagining it. It’s not normal for a society like ours to do that to vulnerable people.*


For these reasons, participants conclude that they often struggle with inconsistencies in healthcare settings, where “scent-free” policies are undermined by the presence of products like hand sanitizers that can act as triggers of symptoms. While some providers are accommodating when concerns are raised, the recurring need to explain and advocate for accommodations demonstrates a lack of consideration and empathy from service providers at essential environments.

#### 3.3.2. Theme 2: Healthcare and Institutional Gaps Influencing Misconceptions

Subtheme 2.1: Insufficient training and awareness among healthcare providers

Participants noted the lack of training in healthcare institutions and the apparent lack of agreement on the validity of MCS as a medical condition. This disconnect often leads them to seek expensive alternative treatments, such as consulting naturopaths or osteopaths. However, participants highlighted the absence of communication and coordination between family doctors and these alternative care providers, further complicating their care journey.


*I do sort of have a doctor. I have [Dr.] from integrated chronic care, [city]. I was fortunate enough to get into his clinic in 2019, just shortly after I collapsed at work, actually, they had an opening. It’s because the clinic is a five day long session you go for. And they put me on a waitlist. I got in, and he diagnosed me with MCS. He gave me some ideas in terms, know math protocols to follow, products to use or not to use, where to get better information […]. And I told him that my family doctor thought I was a whackadoodle. So he called my family doctor and set him straight. Now, my family doctor has since said, I think I’ve got about seven other patients that are like this. [….] So his waitlist is like seven years long. […..]. And so that is the only doctor in land Canada that can work with us, unfortunately. But his paperwork, just for him to say, you have MCS. Yes, you do. You’re not crazy. You are sick. There’s something wrong with you. You have no idea what that does for a person. It doesn’t help us get better at all. But it makes you feel like, yeah, there is something really wrong. And I would just like to say it would be nice if you guys could do sessions for naturopath doctors and osteopath doctors, get them on board, because I think they have lots of patients like us, but apparently, we don’t tell our doctors that we’re seeing naturopaths for some reason.*



*The pain is so. I would call it ungodly in terms of breathing issues. I can barely walk very far. Everything is exhausting. I do have a doctor, however, not only does he not help, he makes my life a lot worse than it already is. He believes I have MCS, but he also believes that you can’t become ill from being in the same room with someone that’s wearing perfume. Or he’ll say, if I ask him for a letter of accommodation saying I need a fragrance free environment, he’ll say he can’t do that because other people have a right to wear perfume. It’s not true. But that’s what he believes and so that’s a disservice to me. Like, you go to work every day, you drive a car, you do all your own chores. None of those three things are true. I haven’t done those things in 30 years. But yet in his mind, because he sees me standing, breathing, not in a wheelchair, that my life is just great and I’m doing all those things that I’m not doing, and if I tell him that’s not true, he argues with me and tells me that I do all those things.*



*I have a doctor who is fairly young now, and she believes in the MCS, but I give her more information than what she knows herself or training. So, the research I’ve done. So that’s about it.*


Subtheme 2.2: The role of government in shaping public perceptions of MCS

Patients report feeling invalidated when government entities fail to take a stand on policies that involve reducing exposure to chemicals. This contributes to societal misunderstandings and stigma towards people who speak out about the lack of compliance with such policies. Furthermore, the lack of policies and enforcement addressing accessibility needs—such as guidance on creating fragrance-free environments—creates tangible barriers to essential services, including healthcare.


*I’ve spoken to public health. They have to bring me the vaccine at home and when their health care people show up, they’re still scented, contaminate my home and I can’t breathe and can’t talk for a week after they come to my house. I cannot access the vaccine. Drugstores aren’t accessible for me, since they started putting hand sanitizer everywhere. I cannot go anywhere. No public space, no doctor’s office, no hospital, nothing.*



*I am really pissed off at the province of [ ] because they have created and are continuing to perpetuate these barriers. They supply the natural concept sanitizer to the city for all the city’s public spaces. This is a procurement and supply chain issue that needs to change. I know federally they’re supposedly on this, but how many more of us literally have to die, lose our health, lose our jobs, lose our relationships and lose our homes? Because these two words are not there: fragrance-free.*


#### 3.3.3. Theme 3: Policy Barriers, Compliance, and Resistance

Subtheme 3.1: Gaps in addressing MCS in government policy

Given that MCS is a condition that has only recently gained broader recognition, participants noted significant discrepancies in how public places acknowledge the impact of fragrances on individuals with MCS. For example, participants recounted experiences of attending facilities that claim to provide a “fragrance-free environment”, only to encounter fragranced sanitizers, scented soaps in washrooms, or healthcare workers using perfumed products like shampoos. Participants expressed widespread agreement regarding the ineffectiveness of posters and advertisements in promoting compliance and addressing resistance to policy implementation, emphasizing the need for more effective reinforcement measures to educate the public and convey the seriousness of the issue. Participants identified gaps in addressing MCS in government policy:

[Government Health Organization] *apparently made a report and recommendations on how to implement it. And the [current administration] has basically decided they’re not going to do anything on it right now. And I was trying to get a hold of this by the Freedom of Information Act, and they won’t allow me to do this. So I’ve talked to some MPs who are trying to do this. Anyways, another big issue is bylaws for different things. So local outdoor bylaws for dryer vents, there’s nothing here in Toronto burning, everybody seems to be burning their yard waste.*

Another participant explained: *They put up posters everywhere, and it never worked, because they told me, there’s no one in management who wants to play perfume police. There’s no one who’s qualified to say, you’re wearing perfume, you’ve got to go home.*

Subtheme 3.2: Resistance within institutions

Multiple participants reported experiencing instances where they noticed resistance within institutions that create barriers to change, which often cascades into impacting other sectors of their lives. For example, participants described institutional reluctance to implement or enforce fragrance-free policies, which not only hindered inclusion in healthcare, workplaces, and educational environments but also perpetuated systemic discrimination. Additionally, participants found themselves frequently accommodating others by leaving establishments when others wore fragrances in areas with fragrance-free policies.

One participant illustrated how institutional inertia within medical settings can delay broader systemic change: Doctor told me that out of seven doctors, three said we should do something. The others weren’t interested. So, by starting with medical clinics, we’re taking the side road because if doctors get used to it or if there are clinics that adopt these methods, maybe it’ll be taken more seriously in hospitals.

Another participant noted that *they will not institute a fragrance-free policy or they will not enforce it. It impacts me where I need to remove myself from that. If they want an accommodation letter, it forces me to go to my doctor’s office to get an accommodation letter. And then he’ll tell me that they have a right to wear perfume. So then that fuels like the impact extends to environment after environment...if I can’t access education because of it, then I have to go to my doctor, and then he further discriminates against me. And then when I tell my doctor, you know, look, I’m going to this venue I need accommodation, I’m getting sicker and sicker because I have to be in this environment. He says you’re not sick. So the impact just extends into every avenue of my life, basically, you know what, it may start in this venue, but it just spreads into other venues of my life, you know?*

#### 3.3.4. Theme 4: Commercial Influences and Misleading Practises

An overarching concern among participants in this study was the increasing use of scented products and the numerous advertisements that promoted the misleading benefits of using such products. The problem with such items is the lack of public questioning related to the use of scented products, particularly in relation to the chemicals associated with them and their potential impacts. In addition, the specific chemicals used in scented products are often undisclosed and classified under the term ‘perfume’ [44], highlighting the lack of transparency and disclosure on labels and in protective legislation. The challenge in this is that the impacts of health concerns have a delayed onset, and multinational commercial industries must be challenged before change can be made.

Participants compare this to multinational industries, such as the Tobacco Industry.


*Well, let’s just say it’s like the petrochemical industry, it’s like the tobacco industry. It took decades for people to win against these multinationals. Me, I see it, it’s the perfume multinational, and people are into it, and they like it. So I don’t see that happening for a very long time.*



*When you read an article where air quality testing was done in the city of Los Angeles, but they discovered that more than half of the smog pollutants in Los Angeles were toxic chemicals coming from dryers and washing machines, that’s really serious. But nobody hears that kind of stuff and anyhow, so, yeah, I just wanted to put that out there. It’s like industry has to be held accountable. That’s the only place I think it’s going to stop.*


Subtheme 4.1: Influence of commercial interest on product safety standards—misleading advertising and greenwashing in consumer products

Due to the excessive barriers that products in various essential locations, such as grocery stores or pharmacies, present to participants from the emission of strong scents, and how pervasively such essential household and personal-care products are used, the participants point out the overarching influence that companies have on consumers, and the contribution of advertisements that impact consumerism. The cultural association of fragrance with identity creates resistance to change, as one tends to associate the natural human scent with a scent that needs to be masked with fragrances.


*Because of advertising, chemical companies and the government and there is such an amalgamation of those things of money and politics and power, and this business of advertising that people just get brainwashed. So we are, we are fighting a very huge issue.*



*One day my two little girls were sitting next to me, and I asked them to count the number of fragrance ads during the movie we were watching, just for fun. We counted nineteen. And the most aberrant, of course, was the one where you see the parents, who have a crying baby and at one point, they look at each other, discouraged, and then they each take a bottle of (laundry brand) and sniff it. I was so discouraged. I thought, they are telling people, if you want to calm down when your baby is crying, sniff some (laundry brand) The conspiracy theory that some of my friends have, it’s like what’s in there something that changes your DNA or what?*


Advertising normalizes and promotes chemically laden products, often downplaying or ignoring their potential health risks, making it harder for people to question or resist their use. For example, one participant criticizes the marketing of essential oils as inherently beneficial, particularly when they are mass-produced or poorly sourced. They argue that such products are often chemical-laden and harmful rather than “natural” or safe, as advertised.


*I think the lavender thing is this false notion that it’s an essential oil. Essential oils are good for you. I say to people, the only time essential oil is good for you is if you’ve bought heirloom seeds, the heirloom vault. You are growing it. You’ve got your own completely organic grow house, grow up. You’ve got organic soil, and you’re using your own distilled oil. And then you are taking your Lavender. When it’s done and you’re emulsifying it yourself, then, yes, it might be essential and safe. Otherwise, than that. Especially when you buy at the goddamn dollar store. It is not good for you. It is a chemical. It is poison. Put it away. You’re making yourself ill. Oh, essential oils.*



*Yeah, the product industry, you know, generally has indoctrinated people to feel that fragrances are identity and there’s a lot of people who say this part of my identity, so we have to fight that. That it’s more harmful than it is positive for you. And one I don’t know if everybody watched market place a few weeks ago when they were talking about the PFAS as in cosmetics, the biggest thing that I took from that episode, which you can find online on that CBC was that it said, PFAS can be found in makeup marked as smudge proof, long lasting or waterproof.*


Subtheme 4.2: Increased intensity and pervasiveness of scented products

Many participants noted a significant increase in the use of fragrances across a wide range of products, including those designed to create a certain ambiance and products intended for animals. They also observed that both the intensity and longevity of these fragrances have noticeably increased.


*Yes, the intensity of scents has increased. And you hear commercials last 12 h or so many weeks, or companies are purposefully making the scents to last longer. I even went online and was able to find a list of the 30 best smelling, long lasting laundry detergents of 2024. So that is something that people are looking for so that they don’t have to wash their clothing as often. I don’t know what their purpose is, but some people just enjoy having odours and scents, and they look for the longest lasting ones. So, it has become a bigger problem for us. Yes.*



*They have these plastic grates that are infused with scents. And I went online and researched about them, and they do come. It’s common for them to come in spiced apple scent. And every manufacturer would say, we have two times more than the other competitors. We have ten times more scent packed in here than the other competitors. And it was almost like a competition between manufacturers of how much scent they had packed into these little whatever screens, and a competition in how much longer those scents last than the other manufacturers. So, it’s in everything. And I think another problem, too. This is going off topic a little bit. But what’s permeated the minds of society is that if something does not smell fragranced, therefore it doesn’t smell fresh, it doesn’t smell clean, and therefore it stinks.*


The increase in the pervasiveness of scented products may be a consequence of COVID-19, which led companies to create disinfectants with added fragrances to incentivize consumers. Indeed, the use of sanitizing and disinfecting products dramatically increased during the COVID-19 pandemic [45].


*I just want to reiterate that I noticed a big change in 2022, and I think that’s when all the container ships had been sprayed down and sprayed down, bringing goods over. Things were showing up, and they were really starting to really smell. And the laundry boosters seemed to be a big thing. And the amount of commercials that come on during the day for these boosters and whatever freakish science it is that if you rub your arm or whatever your clothing, after they use these things, it’s going to reactivate the scent. And I thought, there’s too much with that and the disinfectants that stores are using. And then I don’t know why they’re starting to use (product-masking agent) in change rooms now, but they never used to.*



*I would say I’ve definitely noticed the intensity increase dramatically. I noticed it well before COVID like, for quite a few years prior to that. Covid definitely made it worse because I think people were just like going crazy with the sanitizers and stuff like that. And that’s ingrained in people’s mind. I don’t think that’s going to go away anytime soon or at all. But I think it’s maybe more than just intensity. I think it might be the types of scents that are used. I find, too, that a lot of people don’t wear perfume and colognes anymore. There still are plenty that do. But the types of scents that are now in laundry products and hair sprays and shampoos and deodorants, they now smell like perfume.*



*So I am within the (product-masking agent) and all that stuff, the plugins, that has gotten worse because there’s so much more cleaning products being used because of COVID. And I just keep addressing it with people and addressing it, saying, you know, the only smell that’s worse than the bleach you’ve used to clean something is bleach with Ocean breeze plugins on top of it. Like, at least try to stick to one bad smell. […]. But smells have gone up. People have gone up. The commercials drive me nuts.*


Finally, one participant expressed concern about exposure to scented additives in the vapour emitted from vape pens. These products incorporate substances with scented additives and are popular among the younger generation. This suggests that the addition of fragrances is widespread and being continuously incorporated into various products by multiple manufacturers.


*Probably one of the very worst smells out there for me is the scented vapes. Vapes are bad, but then they throw, like, the cherry smell on top. Oh, my God. If kids are walking down the street and vaping, I almost die.*


## 4. Discussion

This qualitative study examined the experiences of a cohort of 38 individuals with MCS as they encountered misconceptions about their condition, as perceived by people without MCS. These misconceptions are shaped by interrelated factors at both the policy and community levels. At the policy level, misconceptions are influenced by resistance within institutions regarding compliance and implementation of fragrance policies, inadequate government recognition, a limited understanding of its etiology, and a lack of medical consensus surrounding MCS. At the community level, individuals with MCS report systemic psychological misattributions due to the variable presentation of MCS symptoms. Additionally, commercial interests—such as misleading advertising, greenwashing, and the pervasive use of scented products—further reinforce social stigma and barriers to appropriate care, support, and recognition.

In the realm of healthcare, misconceptions often result in misdiagnoses, poor medical care, and inadequate treatment [25,46]. Although experts agree on many characteristics of MCS, discrepancies exist and appear to be reflected in healthcare institutions. Indeed, individuals with MCS are often unable to receive the best medical care because they frequently experience skepticism from their healthcare professionals [25]. Furthermore, the stigma and isolation that they face in social settings and workplaces often lead to discrimination, barriers to accommodation, loss of social circles, and even unemployment [47]. For instance, a systematic review of qualitative studies found that individuals with MCS perceive it as significantly impacting their social and occupational functioning, which are vital aspects of psychological health [36]. This suggests that the impact on social and occupational functioning resulting from MCS is likely the primary driver of the mental health outcomes [48]. However, a limitation of these studies is that they predominantly recruited females from the United States and Canada [36]. The findings highlight the need for further research and the development of targeted prevention and intervention strategies.

Aligning with previous studies, this study found that participants were not only dismissed by medical practitioners for their physiological symptoms, by often associating it with psychological manifestations [25,38,46], but also from their community and family. Participants consistently reported the widespread societal perception of MCS as a psychological condition rather than a physiological one. This misattribution, rooted in the invisibility of symptoms and the subjective nature of triggers, leads to frequent dismissal and invalidation by peers, families, and even healthcare professionals.

The psychological misattribution of MCS extends into healthcare systems, creating significant barriers to obtaining adequate treatment. Many participants described how healthcare providers misinterpreted their condition as a mental health issue, recommending psychiatric interventions rather than addressing the physiological basis of their symptoms. Evidence showcasing that those with MCS have poorer mental health can be explained by the fact that painful and debilitating chronic diseases commonly give rise to disorders such as anxiety and depression [25]. Indeed, in this study, participants reported transitioning from being socially active to withdrawing from public life, not out of choice, but as a necessity to avoid exposure to triggers. The isolation imposed by MCS can strip individuals of their sense of self. Loss of independence, whether in daily activities or the ability to navigate public spaces safely, further compounds the emotional toll, leading to frustration, loss, grief, and a diminished quality of life.

Research suggests the development of training modules for mental health providers who work with persons with MCS [38,46]. Gibson et al. [38] suggest that effective communication with healthcare providers about necessary accommodations, such as fragrance-free environments, is crucial for individuals with these conditions. However, the study also noted that while 75% of participants requested special accommodations, only 50% reported that these requests were fulfilled, indicating a gap between the needs and the accommodations provided. Nonetheless, participants in this study report that acknowledgment of their condition by healthcare providers is crucial in promoting a sense of validation. The study by Briones-Vozmediano & Espinar-Ruiz [25] also demonstrates this finding among women in Spain diagnosed with MCS. These discourses can lend credibility to a patient’s description of their symptoms, reinforcing the validation of their illness. This legitimacy not only promotes social recognition but also enables access to benefits such as disability pensions or sick leave [12].

The care of individuals with MCS is further complicated by the limited integration between traditional medical systems and alternative care providers, such as naturopaths and osteopaths, whom patients often seek out due to necessity. One study found that though individuals with MCS often seek out alternative treatments, the financial impact of spending their limited income on these treatments is significant [49]. Participants in this study suggested that greater collaboration between these fields could enhance the delivery of care. However, stigma against alternative treatments within mainstream medicine remains a significant barrier to such collaboration. Research indicates that diagnosing physicians are more likely than non-diagnosing physicians to use a variety of diagnostic tools, including therapeutic applications and psychological evaluation tests, in partnership with patients [50]. The integration of a multidisciplinary healthcare team may offer a promising approach to improving the effectiveness of treatment options for individuals with MCS [51].

Compounding these challenges is the cultural ubiquity of chemical-laden consumer products, including perfumes and cleaning agents, which individuals with MCS often encounter in healthcare and essential service settings. The ongoing need for advocacy within healthcare settings underscores the systemic lack of awareness and accommodations for individuals with MCS. For example, due to the widely accepted consumer culture of fragrances in various personal and household products, individuals with MCS are often limited in accessing essential services, such as entering pharmacies or grocery stores. It stems down to two reasons. First, they would have to leave their protective environment and expose themselves to various chemicals and are presented with hostility due to the acknowledgement of their disability. Indeed, there are discrepancies in public places that acknowledge the impact of fragrances on people with MCS. People with MCS often experience difficulties when attending facilities that claim to offer a “fragrance-free environment”, yet encounter sanitizers that include fragrances, soaps in washrooms, or face healthcare workers who have used scented shampoos [12]. Since we, as a current society, rely so heavily on the sources of fragment-associated exposures, through consumerism dependency, at the policy level, offering a simple solution, such as appropriate product choice for inclusion, is a solution for accessibility.

Furthermore, participants notice that the intensity and pervasiveness of fragranced products have increased over time. Specifically, a cultural shift toward longer-lasting and more potent fragrances is now often promoted as a desirable feature. For example, the inclusion of fragranced disinfectants and laundry boosters during the COVID-19 pandemic exacerbated exposure risks, demonstrating how public health crises can inadvertently reinforce harmful consumer trends [45,52]. In one study, participants experienced greater exposure to disinfectant or sanitizer odours from it entering their living spaces during the pre- to post-pandemic period (*p* < 0.001). Additionally, their satisfaction with in-person medical visits declined significantly from before the pandemic to after (z = −2.048, *p* = 0.04) [5]. Research suggests options to reduce VOC exposures, including using products containing hydrogen peroxide, opting for unscented products, maintaining proper ventilation, opening doors or windows, and leaving the room for at least an hour [2,29].

The lack of a defined pollution exposure threshold that guarantees no harm to health highlights the complexity of addressing environmental factors associated with MCS [12,53]. This challenge is further exacerbated by the unknown independent, synergistic, or additive effects of chemicals present in household and personal care products (many of which are not fully understood until years after their introduction to the market) [54]. Chronic low-level, long-term exposures to such chemicals often involve a prolonged latency period before the onset of disease. This highlights the importance of longitudinal studies and genetic research in exploring the epigenetic changes triggered by chemical exposures, as well as genetic predispositions that may increase susceptibility to MCS [55]. Bridging these knowledge gaps is critical for improving both preventive measures and targeted interventions.

The globally standardized self-administered BREESI and QEESI, designed to assist researchers and clinicians in screening, studying, and evaluating patients with MCS, have been validated across various populations [21,22,48,56,57]. The use of these inventories in clinical settings among healthcare providers for their patients may increase awareness, diagnoses, and eliminate barriers to accessing adequate treatment.

Finally, patients with MCS require guidance and support from healthcare providers for effective self-management [51]. This includes educating patients to make informed decisions and helping their families understand how to provide support. It has been previously recommended that physicians should address comorbid conditions and advocate for necessary accommodations, as MCS is recognized as a disabling condition in Canada [58] protected under the Canadian Human Rights Act (2007), and assist with third-party disability benefit applications if patients are unable to work [12]. For individuals with environmental sensitivities/multiple chemical sensitivity, accommodations include implementing scent-free policies, minimizing chemical use, selecting less toxic products, and providing advance notice of construction or maintenance work [59].

One limitation of this study is related to the participant selection process. Participants were primarily recruited from a membership base, which may have introduced selection bias, as individuals with more direct access to advocacy groups or resources may have been overrepresented. This could limit the diversity of perspectives, especially among individuals who are not affiliated with these organizations or who may not actively seek support.

Additionally, the study relied on self-reported data, which could introduce recall bias or inconsistencies in how participants described their experiences. Although the sample includes individuals from various regions across Canada, the findings may not be fully generalizable to individuals with MCS outside of these regions, or to those who are less engaged with advocacy efforts or online platforms.

Finally, the study focused on individuals who are familiar with and involved in MCS-related issues, which might not capture the experiences of those who are less aware or less vocal about their condition, further limiting the breadth of perspectives gathered.

## 5. Conclusions

This qualitative study examines the experiences of individuals with MCS as they face perceived misconceptions surrounding their condition at the individual, community, and systemic levels. Due to the general public’s indifference to and tolerance of low-level, common chemical exposures, there is a lack of recognition and psychologization of MCS, which perpetuates barriers to accessing essential services, expressing their needs, and living a good quality of life. Addressing misconceptions about MCS requires a multifaceted approach that includes healthcare education and interventions to improve diagnoses, policy reform, and public awareness. Education and awareness among healthcare providers will help to recognize and accommodate MCS as a legitimate physiological condition. Additionally, it is crucial to implement and enforce fragrance-free policies in public spaces and healthcare settings, backed by government-led initiatives that prioritize inclusivity. Ultimately, there is a need to challenge cultural norms surrounding fragranced products through educational campaigns that counter misleading advertising and promote evidence-based perspectives. By addressing these systemic barriers, society can move toward greater validation, understanding, and support for individuals with MCS, mitigating the stigma and isolation they currently face.

## Figures and Tables

**Figure 1 ijerph-22-01383-f001:**
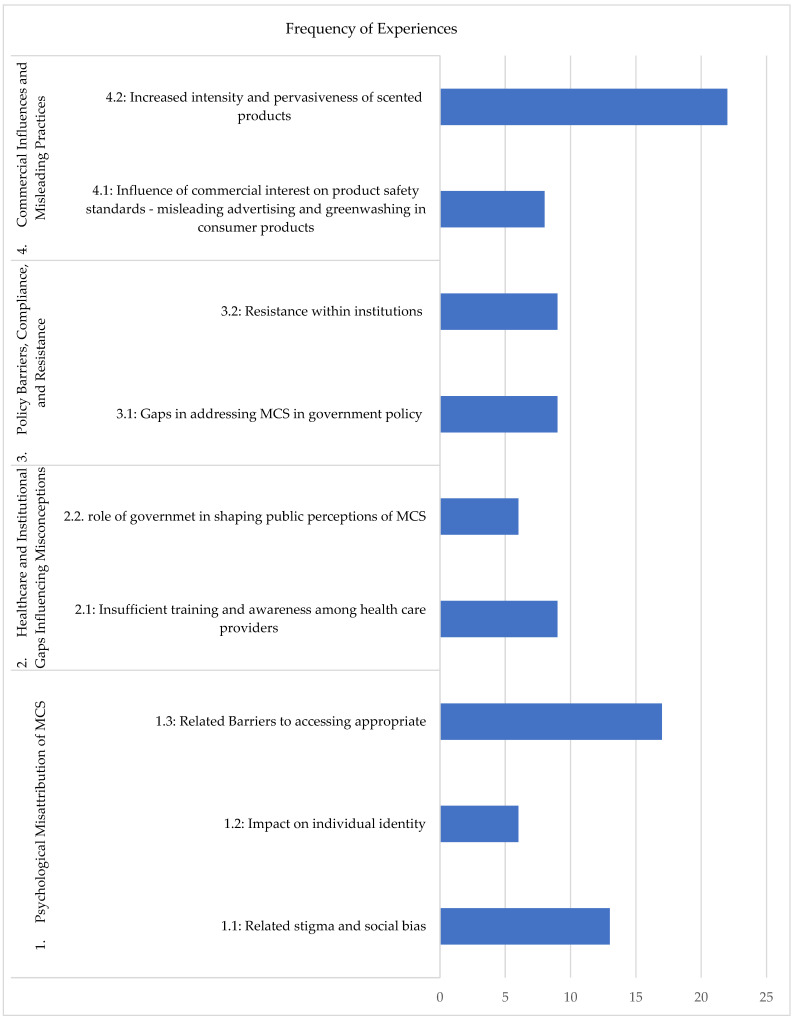
Themes and subthemes related to policy-level and community-level factors as sources of misconceptions surrounding MCS, corresponding to the frequency of experiences shared by participants.

**Table 1 ijerph-22-01383-t001:** Sociodemographic characteristics of participants (*n* = 38).

Demographic Characteristics	*N* (%)
**Age group** **40–49** **50–59** **60–69** **70–79** **80** **Prefer not to answer**	5 (13.2) 14 (36.8) 9 (23.7) 5 (13.2) 2 (5.3) 3 (7.9)
**Gender** **Male** **Female**	4 (10.5) 34 (89.5)
**Location** **Alberta** **British Columbia** **New Brunswick** **Ontario** **Quebec** **United States** **Prefer not to answer**	2 (5.3) 2 (5.3) 1 (2.6) 20 (52.6) 12 (31.6) 1 (2.6) 2 (5.3)

## Data Availability

Data available on request due to restrictions (e.g., privacy, legal or ethical reasons).

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
