# Peer review of "A Qualitative Exploration of Policy, Institutional, and Social Misconceptions Faced by Individuals with Multiple Chemical Sensitivity"

_ijerph, 2025, doi:10.3390/ijerph22091383_

Round 1

Reviewer 1 Report

Comments and Suggestions for Authors

SUMMARY OF MANUSCRIPT

This study investigated the misconceptions of and barriers to seeking medical care for Multiple Chemical Sensitivity (MCS) disease from the perspective of women with the disease. The qualitative study identified four themes: Psychological Misattribution of MCS, Healthcare and Institutional Gaps Influencing Misconceptions, Policy Barriers, Compliance, and Resistance, and Commercial Influences and Misleading Practices. Each theme, with its subthemes, were described and exemplified using quotes from the participants. The discussion situated the results within the literature on MCS, suggested future steps, and addressed limitations of the study.

EVALUATION OF MANUSCRIPT

The manuscript is well written throughout. Though the disease is rare, the consequences of misconceptions and barriers to treatment are very real for these women, and deserves attention in journals such as this one.

The introduction includes the relevant prior research and sets the stage for the current study. The methods are sound, and the data analysis is rigorous. The data presentation is organized and logical. The conclusions follow from the data.

I have a minor request for clarification. On p. 4 at the very bottom, the text says, “Focus groups were conducted by…with individuals who have MCS…on three separate occasions….” The paragraph continues onto p. 5, “Seven focus groups were conducted….” Does this mean that each focus group met three times? Or, were the 7 focus groups collected across the three occasions?

One further comment. The title should more closely tied to the results, something close to the Table 2 title, such as “Policy- and Community-Level Sources of Misconceptions surround MCS.” I am not saying that this should be the title, however. Further, I do not think that the title should include “Not Just in my Head” because it simplifies the results too much.

Author Response

Comment 1: I have a minor request for clarification. On p. 4 at the very bottom, the text says, “Focus groups were conducted by…with individuals who have MCS…on three separate occasions….” The paragraph continues onto p. 5, “Seven focus groups were conducted….” Does this mean that each focus group met three times? Or, were the 7 focus groups collected across the three occasions?

Response: Thank you for pointing this out, we have clarified the information as follows (p. 5, lines 156-158): In total, seven focus groups were conducted across the three occasions, concentrating on lived experiences of avoidance and prevention to address how they intersect with barriers to inclusion and access, and environmental exposure inequities (with particular attention to workplaces and housing).

Comment 2: One further comment. The title should more closely tied to the results, something close to the Table 2 title, such as “Policy- and Community-Level Sources of Misconceptions surround MCS.” I am not saying that this should be the title, however. Further, I do not think that the title should include “Not Just in my Head” because it simplifies the results too much.

Response: As recommended, the title has been revised accordingly: “A Qualitative Exploration of Policy, Institutional, and Social Misconceptions Faced by Individuals with Multiple Chemical Sensitivity”

Reviewer 2 Report

Comments and Suggestions for Authors

The article entitled “Article 1 “Not Just in my Head”: A Qualitative Exploration of Policy, In- 2

stitutional, and Social Misconceptions Faced by Individuals 3 with Multiple Chemical Sensitivity” could be a good article but there is some huge methodologically errors. The main focus of mistakes are related to the type of results and presenting. Also others are:

  1. Title must be more precise and shorter. Please do not use “popular” phrasis for title. It must be scientific and precise. “Not Just in my Head” is maybe good title for project name but for scientific article not.
  2. The aims must be so short precise with high scientific value.
  3. The paragraph “In the realm of healthcare, misconceptions oft ….” From Intorduction should be placed into Discussion section.
  4. Provide a Statistical analysis part.
  5. If it qualitative study, you must present exactly results of study.
  6. Please erase the paragraph :” And I would just like to say it would be nice if you guys could do sessions for naturopath 356 doctors and osteopath doctors, ……..” seems it is narrative. It must be explained in scientific style and mentioned, not fully cited. Also for another subthemes 3.2.2, 3.3., 33.2.
  7. Chole results must be truly reorganized and presented in scientific manner not narrative. Also plese add a results as measures (tables, graphs etc) not only oexplanations
  8. Conclusion must be precise and focused on aims.

Author Response

Comment 1: The article entitled “Article 1 “Not Just in my Head”: A Qualitative Exploration of Policy, In Situtional, and Social Misconceptions Faced by Individuals with Multiple Chemical Sensitivity” could be a good article but there is some huge methodological errors. The main focus of mistakes are related to the type of results and presenting. Also others are:

Response: Thank you for your time in reviewing our manuscript. We have revised the manuscript below as follows, taking in consideration your comments. 

Comment 2: 

The title has been revised accordingly: A Qualitative Exploration of Policy, Institutional, and Social Misconceptions Faced by Individuals with Multiple Chemical Sensitivity

Comment 3: The aims must be so short precise with high scientific value.

Response: 

Thank you for your suggestion, we have clarified the aim of the study to be as follows, please see p. 4, line 138:

“This study aims to explore the policy and community-level misconceptions surrounding MCS, and how these perceptions impact access to healthcare and social inclusion.”

Comment 4: The paragraph “In the realm of healthcare, misconceptions oft ….” From Introduction should be placed into Discussion section.

Response: Thank you for your comment, this paragraph has been moved to the discussion as recommended (Line: 495-505).  

Comment 5: Provide a Statistical analysis part

Response: Thank you for your comment. In terms of the statistical analysis for this study, we have included the relevant statistical methodology for this qualitative study, which involved thematic analysis. Statistical analysis was not applicable due to the qualitative nature of this study. However, rigor was maintained through triangulation, thematic saturation, and coding consensus using NVivo 14. Please refer to page 7, sections 179-191. 

Comment 6: If it is a qualitative study, you must present exactly results of study

Response: The results of the thematic analysis are also presented in Table 2. Please clarify if there any specific details for the analysis that is missing. We believe that we have included all the relevant details of the analysis for this qualitative study.  

Comment 7: 

Please erase the paragraph :” And I would just like to say it would be nice if you guys could do sessions for naturopath 356 doctors and osteopath doctors, ……..” seems it is narrative. It must be explained in scientific style and mentioned, not fully cited. Also for another subthemes 3.2.2, 3.3., 33.2.

Response: 

Thank you for your comment. In terms of the paragraph in question, we believe that it is important to include this direct statement from the participant. This is significant as it highlights the need of participants to seek alternative treatments outside of traditional medical doctors, and the absence of communication and coordination between family doctors and these alternative care providers, which further complicate their care journey (p. 9).

For subtheme 3.2 (“Resistance within institutions”), we have provided further details illustrating the importance of the theme in a scientific style: “Participants pointed to resistance within institutions that creates barriers to change, and that often cascades into impacting other sectors of one’s life. For example, participants described institutional reluctance to implement or enforce fragrance-free policies, which not only hindered inclusion in healthcare and educational environments, but also perpetuated systemic discrimination. One participant illustrated how institutional inertia within medical settings can delay broader systemic change”. (p.11 line 357-360).

Comment: Whole results must be truly reorganized and presented in scientific manner not narrative. Also please add a results as measures (tables, graphs etc) not only explanations

Response: 

We appreciate the reviewer’s feedback and understand the desire for clarity and scientific rigor in data presentation. However, we respectfully note that the current structure of the Results section adheres to published qualitative research methodologies, particularly for thematic analysis.

In qualitative research, it is standard to present themes with interpretative analysis supported by illustrative quotes, allowing the reader to contextualize and understand the lived experiences of participants. The results of the thematic analysis are also presented in Table 2. Please advise if there are specific elements the reviewer feels are missing from the analytic presentation, and we would be pleased to consider further adjustments. We believe that we have included all the relevant details of the analysis for this qualitative study.  

Comment: Conclusion must be precise and focused on aims:

Respone: Thank you for your feedback. We believe that the conclusion highlights the objectives and findings of the manuscript concisely.

Round 2

Reviewer 2 Report

Comments and Suggestions for Authors

The author answered on some questions but the most important results-not. Authors should reorganised the results section, as we previously asked: "Comment: Whole results must be truly reorganized and presented in scientific manner not narrative. Also please add a results as measures (tables, graphs etc) not only explanations"

Also, authenticate result is now 23%, it is not recommended for original research article. 

Author Response

Comment 1: The author answered on some questions but the most important results-not. Authors should reorganised the results section, as we previously asked: "Comment: Whole results must be truly reorganized and presented in scientific manner not narrative. Also please add a results as measures (tables, graphs etc) not only explanations"

Response: We thank the reviewer once again for taking the time to review the revisions made to the manuscript. We respect the reviewer's comments. As recommended, we have represented the most significant findings and themes now in a figure format (please see figure 1), which illustrates the frequency of experiences related to each theme and subthemes. Additionally, we have included a summary of the results, highlighting the most common themes and subthemes in a scientific context. Please see p. 6-7, "Summary of Qualitative Findings". All additions are in red text. 

Comment 2: Also, authenticate result is now 23%, it is not recommended for original research article. 

Response 2: Thank you for bringing this to our attention. We have reviewed the manuscript and improved both the English language and the authentication report, where we analyzed and identified authentication issues. Please see the changes in the tracked changes. 

Round 3

Reviewer 2 Report

Comments and Suggestions for Authors

The authors improved article and answered at almost all comments. Thanks for huge efforts to improve your work. Please take attention on high percent of similar text used (23%), it is recommended to correct before final version.